# The impact of war on HIV/AIDS service provision: In rural health facilities of Tigray, northern Ethiopia, a cross-sectional study

**Migbnesh Gebremedhin Weledegebriel**[1]*, **Haftom Temesgen Abebe**[2], **Kidu Gidey**[3], **Haileselassie Bisrat**[4], **Tekae Gebru**[5], **Niguse Tsegay**[5], **Bisrat Tesfay Abera**[1], **Hailay Gebremeskel**[1], **Demoze Asmerom**[6], **Angesom Gebreweld**[7], **Fikadu Miruts**[7], **Araya Gebreyesus Wasihun**[8], **Kiflom Hagos**[8], **Tesfay Gebregzabher Gebrehiwet**[9]

1 Department of Internal Medicine, College of Health Sciences, Mekelle University, Tigray, Northern Ethiopia, 2 Department of Biostatics, College of Health Sciences, Mekelle University, Tigray, Northern Ethiopia, 3 Department of Clinical Pharmacy, College of Health Science, Mekelle University, Tigray, Northern Ethiopia, 4 Tigray Regional Health Bureau, Tigray, Northern Ethiopia, 5 Department of Pediatrics and Child Health, College of Health Science, Mekelle University, Tigray, Northern Ethiopia, 6 Department of Medicinal Chemistry, College of Health Sciences, Mekelle University, Tigray, Northern Ethiopia, 7 Department of Medical Laboratory Science, College of Health Science, Mekelle University, Tigray, Northern Ethiopia, 8 Department of Medical Microbiology and Immunology, College of Health Science, Mekelle University, Tigray, Northern Ethiopia, 9 Department of Health Systems, College of Health Science, Mekelle University, Tigray, Northern Ethiopia

* migbey12@gmail.com, migbnesh.gebremedhin@mu.edu.et

**Data Availability Statement:** All relevant data are within the paper and its Supporting information files.

## Abstract

### Back ground

HIV/AIDS remained among the common public health problems in developing country. Despite the extensive delivery of ART and improved coverage of the service access, still, man-made problems like war have negatively influenced the utilization of antiretroviral treatment services. The war in Tigray Region in the northern Ethiopia broke out in November 2020 and thereby has brought about an extreme damage on most of the infrastructure in Tigray, including the health institutions. The purpose of this study is, therefore, to assess and report the trend of HIV service provision across the war affected rural health facilities in Tigray.

### Methods

The study was conducted in 33 rural health facilities during the active war in Tigray. A facility based retrospective cross-sectional study design was conducted among health facilities from July 03 to August 05, 2021.

### Result

A total of 33 health facilities from 25 rural districts were included in the HIV service delivery assessment. A total of 3274 and 3298 HIV patients were seen during pre-war period in September and October 2020, respectively. The number of follow-up patients during the war period in January remained to be only 847(25%) which is significantly reduced with a

**Funding:** The author(s) received no specific funding for this work.

**Competing interests:** The authors have declared that no competing interests exist.

**Abbreviations:** ART, antiretroviral treatment; FMOH, Federal Ministry of Health; HIV/AIDS, Human Immune Deficiency virus/Acquired Immune Deficiency State; ICAP, International Center for AIDS Care and Treatment Programs; MSF, Medicine San Frontiers; NGOs, Nongovernmental Organizations; TRHB, Tigray Regional Health Bureau.

P value<0.001. A similar trend was observed during the subsequent months until May. The trend of follow-up patients on ART significantly declined from 1940 in September (pre-war) to 331(16.6%) in May (during the war). This study also revealed a 95.5% reduction of laboratory service provision to HIV/AIDS patients during the war in January and with similar trends thereafter (P<0.001).

## Conclusion

The war has led to a significant decline of HIV service provision in the rural health facilities and most part of the region during the first eight months of active war in Tigray.

## Introduction

HIV/AIDS remained among the common public health problems in developing country. Globally around 37.6 million people were living with HIV by 2020 and of these 27.4 million were enrolled in ART. As of 2020 a total of 745,719 and 65,718 people living with HIV were on ART in Ethiopia and in Tigray region respectively [1]. Data from Tigray Health Bureau before November 2020 indicates that above 65, 718 people living with HIV were getting ART at 141 ART clinic sites across the region [2]. Since the introduction of the antiretroviral therapy (ART), the life span and the quality of life of the infected people have been significantly improved. Despite the extensive delivery of ART and improved coverage of service access, still, man-made problems like war have negatively influenced the utilization of ART services [3]. The brunt of the war on health facilities falls particularly on patients with HIV owing to the disruption of HIV/AIDS programs such as essential services in the prevention of HIV transmission, delivery of basic laboratory services (HIV testing and counseling, viral load and CD4 monitoring), and restricted access to ART [4].

Evidence in war-affected settings reported the destruction of the health system that renders accessibility of health services to the sick and vulnerable mainly evident in patients with communicable diseases [5–7]. A study from sub-Saharan Africa revealed the report of HIV caused deaths due to the grave destruction of the health facilities following the war [7]. Consequently, the war in Tigray Region northern Ethiopia which broke out in November 2020 resulted to looting, vandalizing, as well as extreme damage of health facilities that led to collapse of the health system in the Region [8–10]. A study revealed destruction of healthcare infrastructure and disruption of services: 70% of hospitals, 83% of health center and 712 health posts were non-functional [8]. The aforementioned reports are indications of service impairment of HIV care and complete disruption of the health service in Tigray. Thus, due to the consequences of health service disruption, the increased risk of new infection, complication and death is inevitable. Though there are few studies that assessed the consequences of war on the health system in Tigray, the impact of the war on HIV service delivery has not been assessed yet. Therefore, this study aimed to assess the trend of HIV service provision in terms of HIV patients on follow-up, antiretroviral treatment and laboratory service across the war-affected rural health facilities in Tigray before and during the war periods.

The study will have significant input in assessing the influence of war on HIV service delivery which will be helpful for rehabilitation and enhancing functionality of HIV service in different health facilities of Tigray.

## Method and materials

### Study area

This study was conducted in rural health facilities of Tigray located in the northern Ethiopia. Tigray region comprises 7 zones including one especial zone and 94 districts, with projected population of 6.2 million in 2021 based on the 2007 Ethiopian housing and population census [11]. Before the war, Tigray had a model health system in Ethiopia, with two specialized hospitals (tertiary), 16 general hospitals(secondary), 24 primary hospitals, 224 health centers, 712 health posts, and 269 functional ambulances [2, 10]. In these health facilities, comprehensive HIV care services including diagnosis, treatment and monitoring for 65,718 HIV infected patients were provided [1].

### Study design and population

A facility based retrospective cross-sectional study design was conducted to assess the health service utilization for HIV service delivery assessment from July 03 to August 05, 2021. As ART services are reported to be the highest among other chronic diseases in the healthcare service, patients with HIV were included in the study. To assess the health service status of HIV patients before and during the war; data collectors visited health facilities, retrieve reports and registration books.

### Sample and data source

North West, Central, Eastern, South East and Southern zones were included in the study except Western zone as it was under control of the Ethiopian and Eritrean troops. As majority of the rural setting were highly affected by the war, in this study we have focused in the rural districts. Of 49 rural districts, a total of 25 rural districts which are known to provide minimum required care of HIV services were purposefully selected and included in the study. Of the selected districts, a total of 33 health facilities were selected using systematic random sampling. Reports and registration books were checked over for three pertinent variables a) a follow-up which is conducted to assess the progress of HIV patients, b) their laboratory tests that have been performed and c) if patients receive the ART or not.

### Inclusion and exclusion criteria

From the selected health facilities those who have a complete information were included in the study.

### Data collection procedure and data quality control

A checklist that fits to answer the research question of health service disruption among HIV patients was developed by the research team members. The checklist was conceptualized based on scientific hypothesis which is related to access of healthcare among HIV patients during the war time. The research team members further discussed on the checklist; then, corrections and amendments were made accordingly. It was further customized to the context of the existing situation. Training was given to data collectors and supervisors with health backgrounds. People with different research skills and experiences supervised the data collection process.

Prior to data collection, data collectors and supervisors were reinforced to read and re-read the checklist to be familiarized and get better understanding of the subject matter. The immediate field supervisors play major role in facilitating the overall data collection and ensure the data quality. The completeness and consistency of the data were checked out during supervision.

## Data analysis

The collected data were double entered into a data entry file using Epi Data software, V.3.1. Data were then transferred to SPSS version 20.0 Statistical software for analysis. Descriptive statistics (frequency counts and percentages) were used to summarize the variables.

## Operational definition

**Follow-up patients**: number of HIV patients on follow-up two months before the war erupted and eight months into the war

**Patients on antiretroviral treatment (ART)**: number of HIV patients on ART two months before the war erupted and eight months into the war

**Laboratory service provided**: number of HIV patients who got any of the laboratory service provided in the clinic like CD4, viral load or other basic tests determined; two months before the war erupted and eight months into the war

**Pre-war**: data taken in the month of September and October 2020G.c

**During-war**: data taken during the active war in Tigray, northern Ethiopia starting from November -4-2020 up to June 29–2021

## Ethical Clearance

Ethical Clearance was obtained from the Institutional Review Board of College of Health Science, Mekelle University (Ref.No. MU-IRB 2025/2022). Support letter was obtained from Tigray Regional Health bureau and permission was also granted by selected health facilities before actual data collection. All data were fully anonymized before we accessed them and the ethics committee waived the requirement for informed consent. Confidentiality was maintained at all levels of the study and all the information obtained was used for this study only.

## Results

A total of 33 health facilities from 25 rural districts were included in this HIV service delivery assessment. During the pre-war period (September 2020), there were a total of 3274 HIV patients, of which 63.7% were females. The HIV services; clinical follow-up, laboratory service, and ART provision showed a declining trend during the war period (November 2020 to June 2021) compared with the pre-war September 2020 to October 2020). The clinical follow-up patients of HIV in the month of October 2020 (pre-war) were 2409. However, in the immediate month, that is, when the war broke out, the clinical follow-up of patients dropped to 782 (in the month of November 2020). The number of HIV patients on ART decreased by 83.4% from 1912 in September (pre-war) to 318 in January (war period). Besides, the number of HIV patients on regular follow-up declined from four digits on pre-war (September n = 2409) to two digits during the war period in the months of January (n = 66), February (n = 58), March (n = 93) and April (n = 92). A similar declining trend was observed on the laboratory services, whereby, only 149 HIV patients (14.8%) tested for laboratory during the immediate war period in the month of November compared with 1004 patients who had laboratory test on the pre-war. The laboratory service further showed the reduction of 95.5% (n = 45) service during the month of January compared with the pre-war (Fig 1).

### Trends of HIV service by health facility

During the pre-war period (in the month of September) the clinical follow-up patients of HIV in the sampled health centers and primary hospitals were 1123 and 1293 respectively. The number of patients reduced from 1123 in the month of September to 290 in the month of

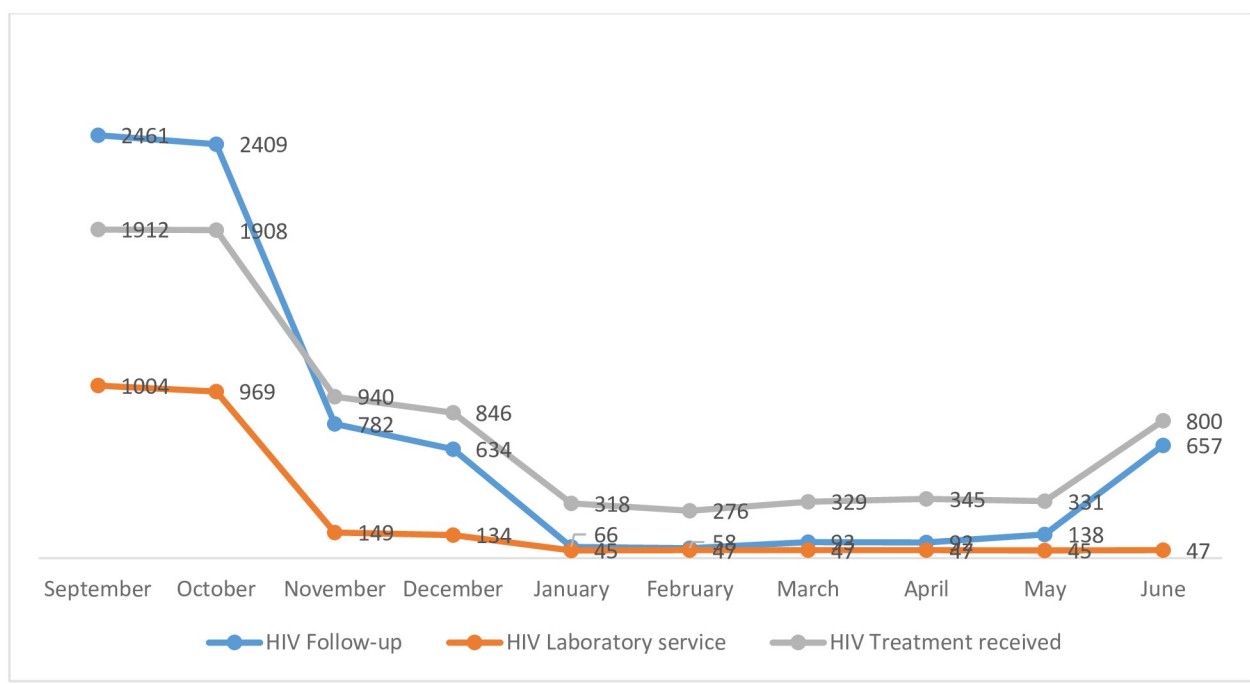

**Fig 1. Trends of HIV follow-up, laboratory service and treatment received before war (September-October) and during the war (Nov-June) in rural Tigray.**

November at health centers and from 1293 pre-war to 492 during the war period in the similar aforementioned months at primary hospitals. In health centers and primary hospitals, the declining trend from four digits (pre-war) to two digits (during the war period) was observed in the month of January. Then, the patients flow continued with similar trend of two digits throughout the remaining four months of the war period (January to May). The number of patients on ART was 755 and 1185 in the month of September (pre-war) at health centers, and primary hospitals respectively. Whereas the counts of the patients reduced to 448 at health centers and 492 at primary hospitals during the first month of the war outbreak showing reduction of 40.7% and 58.5% respectively. In the sampled primary hospitals, the decline of patients on ART was observed from 1196 pre-war (in the month of October) to 83 during the war period (in the month of January) representing 93% reduction. The reduction continued with similar trend throughout the consecutive five months of the war period (January to May 2021) (Fig 2).

## Trends of HIV service by zones in Tigray

Based on the analysis, the trend of HIV service provision was found to be varied among the zones of Tigray Region. A sharp decline of HIV services was observed in the Eastern and Central zones during the war period in comparison to the other three zones. Compared with the first month of the pre-war period (September 2020), the flow of patients with clinical follow-up during the immediate month of the war (in the November 2020) was 0 of 521 (0%) for North west zone, 426 of 1250 (28.2%) for Central zone, 426 of 1250 (34.1%) for Eastern zone, 113 of 230 (49.1%) for South East zone and 170 of 201 (84.6%) for South zone (Fig 3).

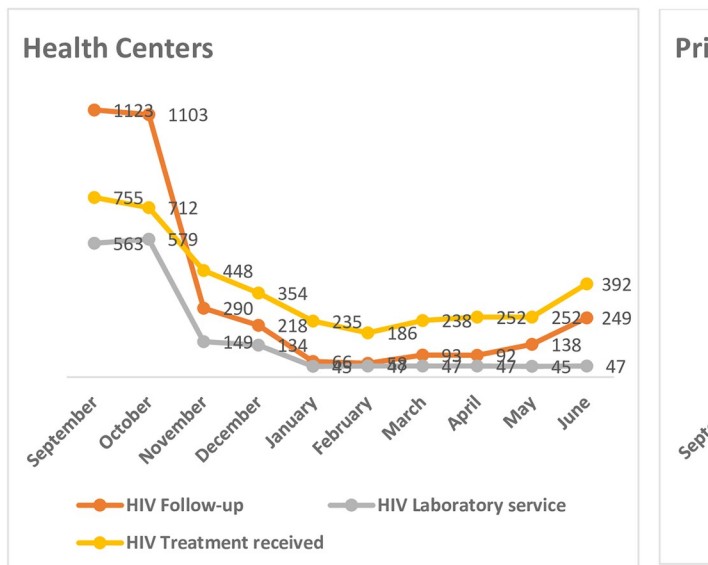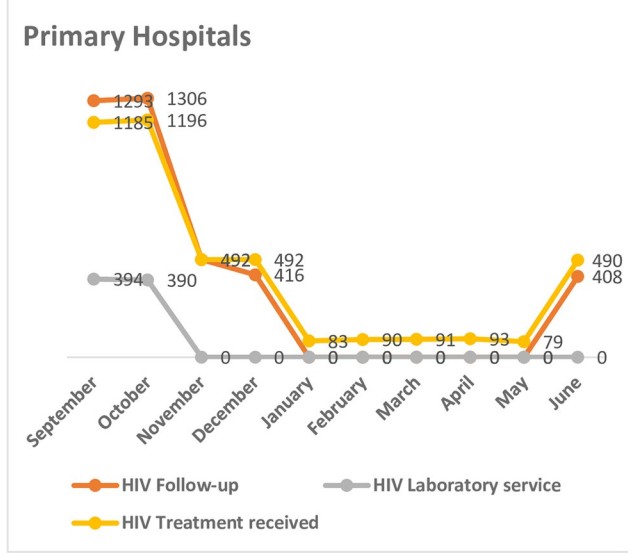

**Fig 2. Trends of HIV follow up, lab service and treatment received before war (September and October) and during war (November-June) for health centers and primary hospitals in rural Tigray.**

## Discussion

In this study, the service delivery of HIV patients was critically hindered by the crisis of the war at the health facilities in the rural districts of Tigray. During the eight months of the war period, the number of HIV patients on clinical follow-up, on antiretroviral drugs, and patients receiving laboratory services were significantly reduced compared with the pre-war period.

The trend of the HIV services over the study period from September to June has significantly reduced across all months compared with the pre-war period. This finding is similar in many war- torn countries like South Sudan, Somalia, Afghanistan, Democratic Republic of Congo, and others where ongoing war led to forced closure of treatment facilities that result in complete or partial stock-outs of ARTs and other essential commodities for HIV service provision and collapses of HIV prevention and control programs [7, 12]. Another possible explanation for the high level reduction of patient follow-up is that the fear of being killed by warring forces on the way to health facility (feeling insecurity), lack of transportation, perceived understanding on looting and vandalizing of health facilities [8, 10]. In contrast to this study a report showed only 25% HIV service reduction during the active conflict in Central African Republic (CAR) in 2016. A similar article also reported different finding from our study whereby 98% of lost to follow-up patients during the conflict outbreak returned back to treatment later [13]. This difference might be explained by the active engagement of NGOs to reach out the most vulnerable groups in unstable community. For instance, there was an effort from MSF in creating a help line communications with patients to solve the challenges of drug shortage through delivering of ART drugs to patients for 3–4 months when active war erupted. However, in Tigray it was difficult to reach out people because of the communication blackout and lack of helpline communication that could have been created by humanitarian agencies.

This study also revealed that the total number of patients receiving antiretroviral drugs decreased by 83.6% in January 2021 and a similar trend was seen in the remaining months of the study period. In accordance with the present results, previous studies in Sudan in 2015

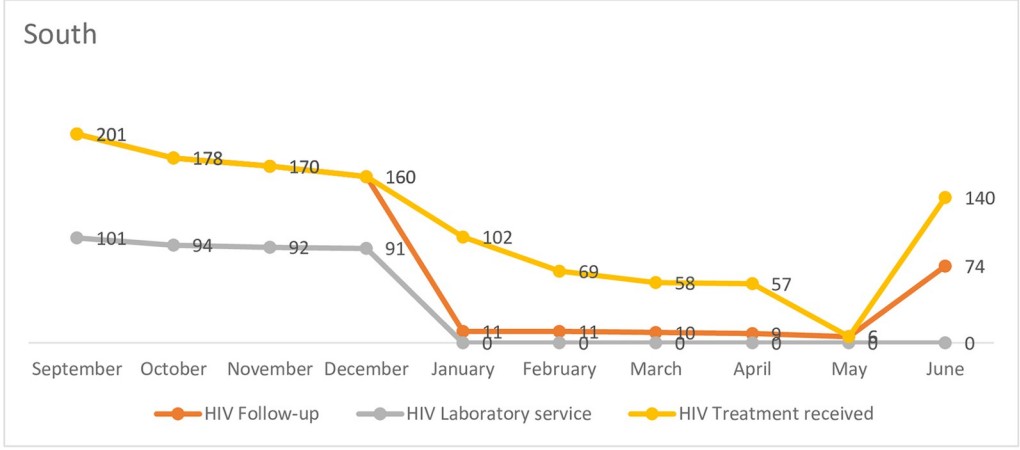

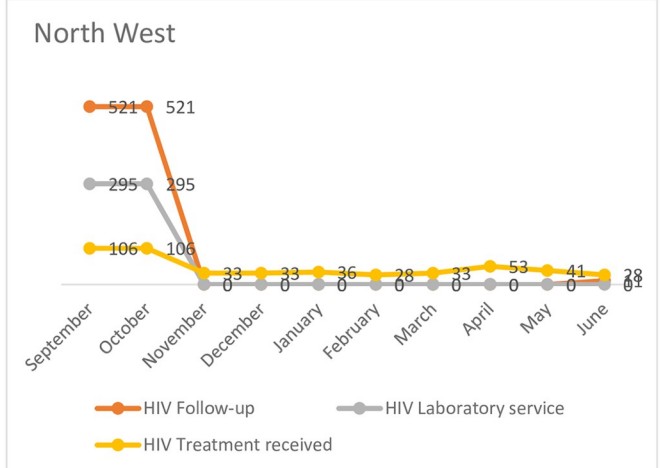

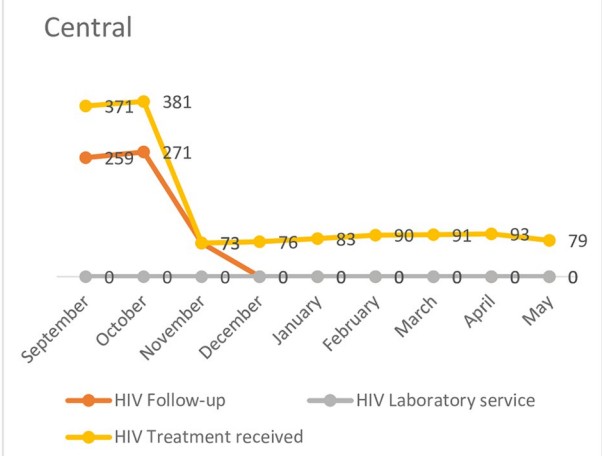

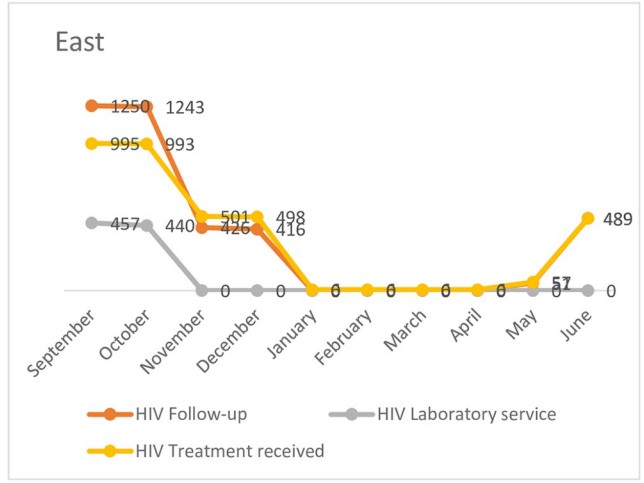

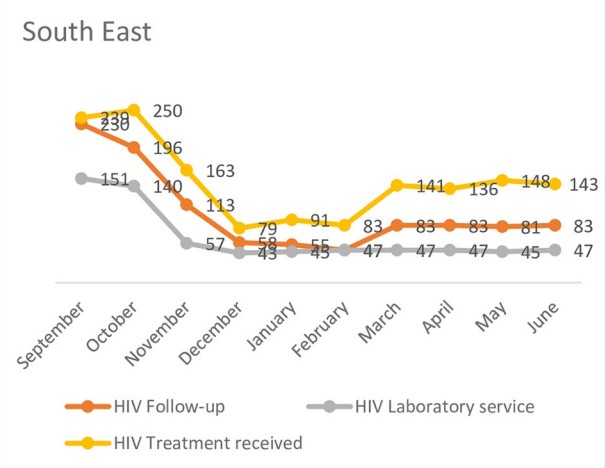

**Fig 3. Trends of HIV follow up, lab service and treatment received before war (September-October) and during the war period by zones (November-June) in rural Tigray.**

conflict have demonstrated that only 10% of eligible adults and 5% of children had access to antiretroviral therapy (ART). Despite the low coverage in South Sudan, it showed a cumulative increase in number of patients enrolled in HIV care and initiated on ART from 2013 up to 2015; this was achieved with the support of ICAP to local partners [12, 14]. The low proportion of patients remaining on ART in Tigray during the war could be as a result of the complete collapse of Tigray's health system following the widespread and deliberate looting, vandalizing, using as settlement center for the army, as well as total destruction of health facilities [10]. Another possible explanation could be the episodes of insecurity created in conflict-affected communities that destabilize and put patients to an increased ART interruptions and drug resistance [15].

Another important finding was that the trend of laboratory service provision to HIV/AIDS clients during the war was dropped by 95.5% in January compared with pre-war period (Sep, 2020). Similar finding was found in Coˆte d'Ivoire during the conflict in 2002 that efforts of prevention and care for people living with HIV/AIDS were significantly reduced with lack of equipment, diagnostic kits and ART drugs were prominent features compared to the period preceding the conflict [16]. Another report from south Sudan seconded this finding where viral load testing was not available in 2015 [14]. In Tigray war, similar conditions that contributed to loss from follow- ups and treatment dropouts could have contributed in addition to the critical shortage of laboratory reagents in the Region.

The results of this study have important implications. The high rate of loss to follow-up and ART discontinuation during the war period in Tigray demands a special attention. Optimal adherence to an ART is critical in determining the efficacy and durability of the regimens [17]. Suboptimal adherence and loss to follow up from care of ART program increases the risk of transmission, disease progression, ART regimen resistance and mortality [18]. In a study conducted in Sudan AIDS related deaths were recorded to be the second highest cause of mortality for internally displaced peoples due to a lack of comprehensive services [12]. Thus, due to the devastating effect of HIV on the individual and community level, it is important that humanitarian assistance programs provide a minimum set of HIV service with active tracing to retain patients on treatment during and after crisis situations. The laboratory service in Tigray was also unavailable for most of the patients. Improvement of laboratory capacity for HIV related services during and after the war is crucial to monitor the patients' status and identify new infections.

Finally, some important limitations need to be considered. The data were extracted retrospectively from reports and registration books that made it difficult to include some other important factors such as treatment for opportunistic infections and risk factors. We were also unable to assess the impact of the service on mortality, drug resistance and other clinical outcomes, since there were no follow-up data for most of the patients. Thus, the effects of the war on clinical outcomes of the disease remain to be investigated.

## Conclusion

In conclusion, the war has led to a significant decline of HIV service provision in the rural health facilities and most part of the region during the first eight months of active war in Tigray. Hence, government bodies and nongovernmental organizations (NGOs) should design strategies to help HIV patients resume their medical follow-ups and proper antiretroviral regimens. Laboratory equipment and reagents necessary for the care of HIV patients should be restored. Resumption of full package HIV care programs in all ART centers with more emphasis to the most affected health facilities should be a priority by Federal Ministry of Health (FMOH), NGOs and other stakeholders.

## Supporting information

**S1 Checklist. STROBE statement—Checklist of items that should be included in reports of observational studies.**
(DOCX)

## Acknowledgments

The authors would like to thank the College of Health Science, Mekelle University and TRHB for creating conducive atmosphere for collecting the data. The authors would like to thank for all staff members of Mekelle University and TRHB who participated in the data collection. We would also like to extend our acknowledgments for the staff members of the health center and primary hospitals for their cooperation in the study. Last but not least we would like to thank Dr.Tesfay Mesele for language editing of the manuscript.

## Author Contributions

**Conceptualization:** Migbnesh Gebremedhin Weledegebriel, Haftom Temesgen Abebe, Kidu Gidey, Tesfay Gebregzabher Gebrehiwet.

**Formal analysis:** Haftom Temesgen Abebe, Haileselassie Bisrat, Araya Gebreyesus Wasihun.

**Methodology:** Haftom Temesgen Abebe, Angesom Gebreweld, Fikadu Miruts, Kiflom Hagos, Tesfay Gebregzabher Gebrehiwet.

**Project administration:** Migbnesh Gebremedhin Weledegebriel.

**Supervision:** Migbnesh Gebremedhin Weledegebriel, Haileselassie Bisrat, Tesfay Gebregzabher Gebrehiwet.

**Visualization:** Migbnesh Gebremedhin Weledegebriel, Tekae Gebru, Niguse Tsegay, Bisrat Tesfay Abera, Hailay Gebremeskel, Demoze Asmerom.

**Writing – original draft:** Migbnesh Gebremedhin Weledegebriel.

**Writing – review & editing:** Migbnesh Gebremedhin Weledegebriel, Haftom Temesgen Abebe, Kidu Gidey, Haileselassie Bisrat, Tekae Gebru, Niguse Tsegay, Bisrat Tesfay Abera, Hailay Gebremeskel, Demoze Asmerom, Araya Gebreyesus Wasihun, Tesfay Gebregzabher Gebrehiwet.

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
