## [Decision Letter · Decision Letter 0]

17 Feb 2023

PONE-D-22-32611The Impact of War on HIV/AIDS Service Provision: in Rural Health Facilities of Tigray, Northern EthiopiaPLOS ONE

Dear Dr. <table class="NormalTable"> 

Welegebriel

</table>

Thank you for submitting your manuscript to PLOS ONE. After careful consideration, we feel that it has merit but does not fully meet PLOS ONE’s publication criteria as it currently stands. Therefore, we invite you to submit a revised version of the manuscript that addresses the points raised during the review process.

-Line number 71: As it is difficult to know whether it is deliberate or not, I suggest to delete the word " deliberate" from the sentence.

Justify why rural was selected 

-Did the study include distracted (partial or complete) health facilities? Was there any assessment before the study about the status of health facility before random selection? 

-Why two months (September and October) were selected as pre-war period? 

Line number 208: there is some improvement in communication, revise the statement. 

We look forward to receiving your revised manuscript.

Kind regards,

Musa Mohammed Ali, PhD

Academic Editor

PLOS ONE

Journal Requirements:

3. We note that you have stated that you will provide repository information for your data at acceptance. Should your manuscript be accepted for publication, we will hold it until you provide the relevant accession numbers or DOIs necessary to access your data. If you wish to make changes to your Data Availability statement, please describe these changes in your cover letter and we will update your Data Availability statement to reflect the information you provide

Reviewers' comments:

Reviewer's Responses to Questions

**Comments to the Author**

1. Is the manuscript technically sound, and do the data support the conclusions?

Reviewer #1: Partly

Reviewer #2: No

2. Has the statistical analysis been performed appropriately and rigorously? 

Reviewer #1: No

Reviewer #2: Yes

3. Have the authors made all data underlying the findings in their manuscript fully available?

Reviewer #1: Yes

Reviewer #2: Yes

4. Is the manuscript presented in an intelligible fashion and written in standard English?

Reviewer #1: Yes

Reviewer #2: Yes

5. Review Comments to the Author

Reviewer #1: Abstract:

Line 31. is HIV problem for all the world? Make some specific phrasings like in developing country.

Line 51. Key word ART and treatment are same (better to delate duplication)

Line 54. Same comment to line 31

Line 56. Your reference for ‘The impact of HIV is highly pronounced in war-affected settings, including Tigray, northern Ethiopia.’

Methodology

Line 104-105 why purposive selection of 25 out of 84 how to be sure that these 25 represent the rest 84?

Again, in line 106-107, how random selection of 33 can give the image of the whole health center unless systematic sampling made?

Line 111. Do the selected health facilities have complete information for inclusion? If not how much…

Line 129 I don’t think the operational definition is important.

Result and discussion

All the data was analyzed in so simple calculator no Stastically analyzed which make it not a research article so, I suggest better to be short report rather than full research article.

Reviewer #2: This is a study of great importance highlighting the impact of war on health care delivery and I applaud your effort you made to quantify and publish the suffering. here are my important questions:

1. The study included 33 rural facilities in Tigray. They have mentioned that there are 84 districs and of those 25 districs were selected . what was the criteria to select those districts? Why not look at the data in all 84 districts or districts from which they can get information? Would this introduce bias by either including those they have access to or those that are heavily damaged?

2. There were 141 ART clinics pre war- how many clinics did they include in their study?

3. 65,718 HIV patients were receiving care prewar. In the health facilities studied, only 3274 patients were seen pre war making this 5% of the total patient population who were getting care. Unless we know and justify why facilities who were providing only5% of care were included in this study, the result can not be informative and generalizable to represent what happened to Tigray HIV care.

4. They looked at the data from November 2020 to June 2021. The war has been going on until December of 2022 and the damage, siege and deprivation was extreme after June of 2021. Why did they decide to review the first six month only. This clearly simplifies the challenge and extent of damage and disruption of care.

6. PLOS authors have the option to publish the peer review history of their article (what does this mean?). If published, this will include your full peer review and any attached files.

Reviewer #1: No

Reviewer #2: No

---

## [Author Response · Author response to Decision Letter 0]

5 Apr 2023

Dear Prof. Musa Mohammed,

First of all, we would like to thank you very much for the chance you gave us to revise the manuscript. We were very pleased to know that our manuscript was rated as potentially acceptable for publication in PLOS, subject to a radical revision and response to the comments raised by the two referees.

Based on the instructions provided in the journal's website and your email on Feb 17, 2023, we attached the file of our revised manuscript in word (doc.). We have revised the manuscript along the line of all comments made by the two referees. 

Appended to this letter is our point-by-point response to the comments raised by the two referees and editor. 

Sincerely,

Migbnesh

Response to reviewers and editor 

First of all, we would like to thank you very much for the comments.

Comments from Editor

-Line number 71: As it is difficult to know whether it is deliberate or not, I suggest to delete the word " deliberate" from the sentence.

Response:

Thank you for the comments. As of your suggestion we have deleted the word “deliberate”. Please see the revised manuscript with track changes.

-Justify why rural was selected

Response:

Thank you again. The reason why we select the rural setting is based on our research objective. Our objective was to assess and report the trend of HIV service provision across the war affected rural health facilities in Tigray. Majority of the rural setting in Tigray were highly affected by the war. We have now mentioned this in the revised manuscript. Please see the revised manuscript with track changes. 

-Did the study include distracted (partial or complete) health facilities? Was there any assessment before the study about the status of health facility before random selection?

Response:

Yes, we have included all the selected health facilities regardless of their distraction. We didn’t conduct an assessment before the study about the status of the health facilities. Please see the revised manuscript with track changes.

-Why two months (September and October) were selected as pre-war period? 

Response:

To make the comparison of the functionality of the health facilities before the war and during the war period. In order to assess the impact of the war on health service provision taking the immediate months before the war versus during the war period would make the comparison reasonable.

-Line number 208: there is some improvement in communication, revise the statement. 

Response:

Thank you we accept your suggestion. The statement on line 207 to 213 is revised and improved. Please see the revised manuscript with track changes

Reviewer 1

Abstract: 

Line 31. is HIV problem for all the world? Make some specific phrasings like in developing country.

Response: 

Thank you As per your suggestion we have now modified this. Please see the abstract in the revised manuscript with track changes. 

Line 51. Key word ART and treatment are same (better to delate duplication)

Response:

Thank you again. We agree. Please see the revised manuscript with track changes. 

Line 54. Same comment to line 31

Response: 

We have done this. Please see the revised manuscript with track changes. 

Line 56. Your reference for ‘The impact of HIV is highly pronounced in war-affected settings, including Tigray, northern Ethiopia.’ 

Response:

Thank you for the comment. This sentence is removed from the text

Methodology 

Line 104-105 why purposive selection of 25 out of 84 how to be sure that these 25 represent the rest 84? 

Response:

Thank you for raising a critical comment and with due respect we have made a correction for the number 84 to 49 districts, this was a mistake during the write-up (look at the manuscript with track changes). As stated on the method section; purposive selection of the districts was made by design in order to capture the eligible health facilities engaged in the provision of the minimum required standard of health care of HIV services. From the onset, the implementation of ART was introduced as part of strengthening the health system, whereby, districts are often selected by purpose to be ART sites based on the magnitude of the cases. The districts with small number of people living with HIV (PLHIV) were not selected since they don’t represent for the total number of ART beneficiaries in all the districts. 

Regarding the issue of representativeness; as World Health Organization (WHO) recommended the minimum number of representative premises (clusters) is 30%, out of 49 districts we took 25 districts which represents 50%.

Again, in line 106-107, how random selection of 33 can give the image of the whole health center unless systematic sampling made?

Response:

Thank you. The comment is well accepted. Indeed, the 33 health facilities were selected using systematic random sampling. We have now mentioned this in the manuscript. Please see the revised manuscript with track changes

Line 111. Do the selected health facilities have complete information for inclusion? If not how much…

Response:

Yes, all the selected health facilities had complete information for inclusion. 

Line 129 I don’t think the operational definition is important.

Response:

Thanks for your suggestion. We could omit the operational definition from the manuscript; however, we believe that the definition might give an insight for our readers.

Result and discussion 

All the data was analyzed in so simple calculator no Stastically analysed which make it not a research article so, I suggest better to be short report rather than full research article.

Response:

Thank you for your suggestion. Statistical analysis can be descriptive or analytical. In this study we used a descriptive statistical analysis that answer our research objective. For example, we have used different trends to compare the results between pre-war and during war period. Therefore, we believe that the statistical analysis used in this research is acceptable in answering our research objective.

Reviewer 2:

Reviewer #2: This is a study of great importance highlighting the impact of war on health care delivery and I applaud your effort you made to quantify and publish the suffering. here are my important questions:

1. The study included 33 rural facilities in Tigray. They have mentioned that there are 84 districs and of those 25 districs were selected. what was the criteria to select those districts? Why not look at the data in all 84 districts or districts from which they can get information? Would this introduce bias by either including those they have access to or those that are heavily damaged?

Response: 

Thank you for the comments and with due respect we have made a correction for the number 84 to 49 districts, this was a mistake during the write-up (look at the manuscript with track changes). As World Health Organization (WHO) recommended the minimum number of representative premises (clusters) is 30%, out of 49 districts we took 50% (25) districts which provide minimum standard health care of HIV services. Capturing the data from all (49) districts was not feasible in cost wise. 

Given the systematic random sampling technique was used to select the 33 health facilities from the purposively selected 25 districts, we believe it couldn’t create a bias. 

2. There were 141 ART clinics pre war- how many clinics did they include in their study?

Response:

We included 33 health facilities (ART clinics) in this study.

3. 65,718 HIV patients were receiving care prewar. In the health facilities studied, only 3274 patients were seen pre war making this 5% of the total patient population who were getting care. Unless we know and justify why facilities who were providing only5% of care were included in this study, the result cannot be informative and generalizable to represent what happened to Tigray HIV care.

Response: 

Thank you again for the comments. 

Our objective was to assess the service which was provided in the primary health care units where majority of the primary hospitals and health centers are located in the rural setting. We selected the specific ART clinics because the war was undergoing in the rural setting. Therefore, we believe that the selected facilities will represent the primary health care units of the rural setting in Tigray and it will not affect the generalizability.

4. They looked at the data from November 2020 to June 2021. The war has been going on until December of 2022 and the damage, siege and deprivation was extreme after June of 2021. Why did they decide to review the first six month only. This clearly simplifies the challenge and extent of damage and disruption of care.

Response: 

Thank you for the comment. After relative peace resumes on June 2021, we had an access to conduct the survey and we never expected continuation of war and siege. The continuation of war phases and the siege crises were another scenario that could have been assessed later.

---

## [Decision Letter · Decision Letter 1]

16 Apr 2023

The Impact of War on HIV/AIDS Service Provision: in Rural Health Facilities of Tigray, northern Ethiopia, a Cross-sectional Study

PONE-D-22-32611R1

Dear Dr. Weledegebriel

We’re pleased to inform you that your manuscript has been judged scientifically suitable for publication and will be formally accepted for publication once it meets all outstanding technical requirements.

Kind regards,

Musa Mohammed Ali, PhD

Academic Editor

PLOS ONE

Additional Editor Comments (optional):

Reviewers' comments:

Reviewer's Responses to Questions

**Comments to the Author**

1. If the authors have adequately addressed your comments raised in a previous round of review and you feel that this manuscript is now acceptable for publication, you may indicate that here to bypass the “Comments to the Author” section, enter your conflict of interest statement in the “Confidential to Editor” section, and submit your "Accept" recommendation.

Reviewer #1: All comments have been addressed

Reviewer #2: (No Response)

2. Is the manuscript technically sound, and do the data support the conclusions?

Reviewer #1: Yes

Reviewer #2: Partly

3. Has the statistical analysis been performed appropriately and rigorously? 

Reviewer #1: Yes

Reviewer #2: No

4. Have the authors made all data underlying the findings in their manuscript fully available?

Reviewer #1: Yes

Reviewer #2: Yes

5. Is the manuscript presented in an intelligible fashion and written in standard English?

Reviewer #1: Yes

Reviewer #2: Yes

6. Review Comments to the Author

Reviewer #1: I can say all my first comments and questions were amended correctly. So my saying is to accept for publication.

Reviewer #2: (No Response)

7. PLOS authors have the option to publish the peer review history of their article (what does this mean?). If published, this will include your full peer review and any attached files.

Reviewer #1: **Yes: **MENGISTU Hailemariam zenebe

Reviewer #2: No

---

## [Editor Report · Acceptance letter]

24 Apr 2023

PONE-D-22-32611R1 

The Impact of War on HIV/AIDS Service Provision: in Rural Health Facilities of Tigray, northern Ethiopia, a Cross-sectional Study 

Dear Dr. Gebremedhin Weledegebriel:

I'm pleased to inform you that your manuscript has been deemed suitable for publication in PLOS ONE. Congratulations! Your manuscript is now with our production department. 

Kind regards, 

on behalf of

Dr. Musa Mohammed Ali 

Academic Editor

PLOS ONE